# OpenReview forum: "Learning Human-like Representations to Enable Learning Human Values"
_NeurIPS.cc/2024/Conference — NeurIPS 2024 poster_

### Official Review · Reviewer_RqUK · 2024-06-13

**Soundness:** 2
**Presentation:** 3
**Contribution:** 3
**Rating:** 6
**Confidence:** 3

**Summary:**

The paper explores how representational alignment between humans and AI agents affects the ability of AI systems to learn human values efficiently and safely. The authors propose that AI systems learning human-like representations can generalize human values better and ensure safer exploration during learning. They support their hypothesis through theoretical analysis and extensive experiments, including simulations and human value judgments in a reinforcement learning setting.

**Strengths:**

I have a positive view of this work. The paper presents a novel approach by linking representational alignment with value alignment, addressing a significant challenge in AI safety. The experiments are thorough and well-structured, covering various aspects of human values and employing multiple machine learning models.

**Weaknesses:**

1. The theoretical analysis relies on strong assumptions, such as specific kernel functions and Gaussian process regression, which may limit the generalizability of the results. Discuss the impact of these assumptions on real-world applicability and consider additional theoretical or empirical validations to support the generalizability of the method.
2. Using the Spearman correlation coefficient as a measure of alignment might have its advantages. However, it is crucial to investigate whether other alignment metrics have been considered or tested. Conducting comparative experiments to evaluate the impact of different alignment metrics on the experimental results will help in understanding the robustness and reliability of the chosen metric and could potentially identify more effective ways to measure representational alignment.
3. Presentation
(a) I suggest that the authors highlight the best results in the tables and clarify in the caption whether each metric is better when higher (Mean Reward) or lower (Immoral Actions Taken). Additionally, they should note any significant p-value with an asterisk (*). (b) I also suggest that the authors change the colors of the GPT models and Embedding Models in Figure 3, as the current colors are difficult to distinguish at first glance.

**Questions:**

1. Can the authors clarify the relationship with inverse reinforcement learning and imitation learning?
2. Although the authors have demonstrated in the experiments that the method in the paper is indeed effective, one point still puzzles me: in Figure 3, in the right panel (generalization phase), it is evident that with the increase of representational alignment, the mean reward increases and immoral actions taken decrease. The number of immoral actions taken decreases to near zero in the generalization phase at the point near 0.35, which is even better than the same level of representational alignment in the personalization phase. Can you explain the reason for this?

**Limitations:**

1. The findings might not apply to all machine learning models and architectures. Additional experiments with different models and architectures would strengthen the conclusions and demonstrate the broader applicability of the results.
2. The human value judgments used in the experiments were collected from a relatively homogenous group (English-speaking internet users from the US). This limits the generalizability of the findings to a more diverse population. Future studies should include a wider range of participants to ensure that the results are universally applicable.

---

> ### Author Rebuttal · Authors · 2024-08-07
>
> *> The theoretical analysis relies on strong assumptions, such as specific kernel functions and Gaussian process regression, which may limit the generalizability of the results. Discuss the impact of these assumptions on real-world applicability and consider additional theoretical or empirical validations to support the generalizability of the method.*
>
> We appreciate the suggestion to further test the generalizability of our results beyond the kernel-based experiment, and we provide a more realistic implementation of our kernel-based experiment using LLMs. To address this, we have expanded the results from our work by including a more extensive evaluation of large language models performing the text-based learning task, which we have outlined in the general response.
>
> *> Using the Spearman correlation coefficient as a measure of alignment might have its advantages. However, it is crucial to investigate whether other alignment metrics have been considered or tested. Conducting comparative experiments to evaluate the impact of different alignment metrics on the experimental results will help in understanding the robustness and reliability of the chosen metric and could potentially identify more effective ways to measure representational alignment.*
>
> While choosing the Spearman correlation coefficient, we did investigate other possible metrics and compare them. We provide a list of metrics we considered below and why we rejected them:
>
> - Pearson correlation: Spearman correlation is able to capture non-linear relationships, because it is ordinal, whereas Pearson cannot. Individual similarity matrices may be on different scales or have different biases (e.g. tending towards higher or lower ratings), and Spearman correlation enables an equivalent comparison of these matrices regardless of these factors. Our theory section in the paper also supports this choice.
>
> - Spearman correlation between all pairs of (personalization, generalization) actions, personalization actions only, or generalization actions only: This measure is sensitive to the specific choice of personalization vs generalization set, and does not accurately reflect the overall degree of representational alignment between two agents.
>
> We appreciate the reviewer’s comment and will add a section to the appendix describing these alternative metrics.
>
> *> Presentation (a) I suggest that the authors highlight the best results in the tables and clarify in the caption whether each metric is better when higher (Mean Reward) or lower (Immoral Actions Taken). Additionally, they should note any significant p-value with an asterisk (*). (b) I also suggest that the authors change the colors of the GPT models and Embedding Models in Figure 3, as the current colors are difficult to distinguish at first glance.*
>
> We thank the reviewer for their helpful suggestions to improve clarity and will implement these changes in the final version of the paper.
>
> *> Can the authors clarify the relationship with inverse reinforcement learning and imitation learning?*
>
> Inverse reinforcement learning explicitly models the reward function of the demonstrator and seeks to infer it from their actions. Imitation learning uses the actions of the demonstrator in specific states and tries to learn that function directly. Both are different from our setting, in which the agent simply performs a reinforcement learning task and receives feedback on the actions it takes based on human values. The agent has no explicit representation of the reward function of the human or the actions they would take, but is trying to learn good actions in an “environment” created by a human’s values. We will clarify this distinction in the final version of the paper.
>
> *> Although the authors have demonstrated in the experiments that the method in the paper is indeed effective, one point still puzzles me: in Figure 3, in the right panel (generalization phase), it is evident that with the increase of representational alignment, the mean reward increases and immoral actions taken decrease. The number of immoral actions taken decreases to near zero in the generalization phase at the point near 0.35, which is even better than the same level of representational alignment in the personalization phase. Can you explain the reason for this?*
>
> In the experiment shown in Figure 3, during the personalization phase, all agents are in the process of learning to identify moral and immoral actions. This means that they will almost certainly take at least some immoral actions during their learning process, because they start with only their representation space and no other knowledge of the actions or their rewards. However, in the generalization phase, the agents have already gone through the personalization phase, during which they learn what actions are moral vs immoral. This results in the observation you have made, that the agents that have higher representational alignment are able to generalize their learnings exceptionally well to new, unseen actions (hence, near-zero immoral actions taken).

---

> > ### Comment · Reviewer_RqUK · 2024-08-10
> > **Thank you for the responses!**
> >
> > Thank you to the authors for their responses. Most of my questions have been addressed. After considering your responses and the feedback from other reviewers, I will maintain my evaluation.

---

> > > ### Author Response · Authors · 2024-08-12
> > >
> > > Thank you for your response!
> > >
> > > If you do have any remaining concerns, please let us know, and we'd be happy to try to address them!

---

> ### Author Response · Authors · 2024-08-07
>
> *> The findings might not apply to all machine learning models and architectures. Additional experiments with different models and architectures would strengthen the conclusions and demonstrate the broader applicability of the results.*
>
> We used three different kernel-based methods (kernel regression, support vector regression, and Gaussian process regression) to produce our results in the paper, and in the experiment using human morality judgments, we used similarity kernels obtained from a variety of embedding models via distance measures between embeddings as well as from multiple LLMs via prompting for similarity judgments. We have additionally further expanded the results from our work by including a more extensive evaluation of large language models performing the text-based learning task, which we have outlined in the general response. We appreciate the reviewer’s suggestion to demonstrate broader applicability of our results.
>
> *> The human value judgments used in the experiments were collected from a relatively homogenous group (English-speaking internet users from the US). This limits the generalizability of the findings to a more diverse population. Future studies should include a wider range of participants to ensure that the results are universally applicable.*
>
> We appreciate the reviewer’s suggestion and have called out this limitation of our work in the discussion section of our paper. We believe that future studies would benefit from a wider and more representative group of humans providing value judgments.

---

### Official Review · Reviewer_7UgR · 2024-07-12

**Soundness:** 3
**Presentation:** 2
**Contribution:** 2
**Rating:** 5
**Confidence:** 4

**Summary:**

The authors test whether human alignment of LLM representations are related how well LLMs can learn personalised preferences. They collect preference ratings and similarity judgements from humans for various value-related stimuli. The authors use the ratings to construct a reinforcement learning problem for LLMs. Using both simulations and the human data collected, they show that LLMs whose representations’s kernel align with the kernel of the human similarity judgements gain more reward (i.e. cater better for human preferences) and select fewer unsafe actions. While the initial results presented are in the domain of morality, the authors show these results generalise to other value-based domains such as compassion and fairness.

**Strengths:**

- The work focuses on the timely problem of personalising LLM behaviour in a safe manner.
- The benefit of representational alignment for safe and effective personalisation is shown in several domains and using several function approximators.
- Several LLMs that span both open-source and closed-source space are considered. It is nice to see that the two different ways of obtaining representations from open vs. closed source models yield similar results, which can be beneficial for guiding future work.

**Weaknesses:**

- The hypothesis that increased representational alignment allows for better learning of human values is not actually tested in the paper. If I understand correctly, all the LLM experiments are done using a kernel derived from representations and some kernel-based function approximator. This is not humans interact with LLMs. The paper can be greatly improved by running the exact same bandit experiment with by prompting LLMs and obtaining their behaviour. If the LLMs that have more aligned representations with humans perform better in the behavioural version of the task, the proposed hypothesis would be supported. Otherwise there is no way to know if the current bandit results translate to behaviour. In fact, just the results of such a behavioural experiment alone would be sufficient to test this hypothesis. It is unclear to me what the benefit of the kernel-based reward learning approach is. Would it not suffice to correlate reward obtained from behaviour with the representational alignment?
- The results of the simulations are not informative. What is being shown is if you corrupt the true generative kernel that goes into the function approximator (i.e. decrease representational alignment), the model performs worse. Using the type of corruption you employ, where some scores a randomly assigned, this is to be expected under any function approximation problem with the types of models you use. I think these findings can either go into Appendix or can be removed, as they currently take up a lot of space and attention in the paper.
- The conclusion drawn from the control experiment is confusing. If you mismatch the kernel with the reward function the model needs to learn, the performance goes down during personalisation. However, Table 3 in the Appendix shows that a human kernel performs better than a length kernel when the rewards are defined over length during generalisation. I appreciate the authors discuss the reasons behind this. However, the findings are followed by “[…]human-like representations are not necessarily always helpful for all tasks. Conversely, a representation based on a reward function such as the length of an action description can help support safe exploration in that particular task, but not for learning human values.” It does seem like human representations are helpful in the both tasks you defined, assuming the generalisation phase is more important than the personalisation phase.

- More minor suggestions to improve clarity:
    - It is hard to distinguish the two green colours used in Figure 3. Your legends overlap with the y-axis labels.
    - “The representational alignment of a particular agent is measured as the Spearman correlation between the upper triangular, off-diagonal entries of the corrupted and actual similarity matrix (because diagonal entries are all the same, and the similarity matrix is symmetric).” =-- - - Pretty much the same sentence is repeated later in the text.
   - Please consider making your figure captions more informative. Figure 3 and Figure 4 almost have identical captions. Also, for Figure 3, the caption only mentions embedding models but that is not just what is plotted. Some of the sentences in the main text, such as the one starting on line 280, can be moved to the caption for better flow.

**Questions:**

- I’m not sure what “s.t. |a| = 10.” refers to in the pseudocode.
- I’m confused about how actions are sampled from the function approximators’ estimates. Do you do Thompson sampling from the estimates? Pseudocode suggests so. However, in the text, Thompson sampling is described completely separately

**Limitations:**

I think the authors raise some of the important limitations in the text. In fact, they highlight that a behavioural experiment with LLMs can be useful, but this is discussed with respect to faster convergence. I believe the paper would improve greatly if the weakness points I raised are addressed, especially the first one. Then, I would be happy to consider increasing my score!

---

> ### Author Rebuttal · Authors · 2024-08-07
>
> *> The hypothesis that increased representational alignment allows for better learning of human values is not actually tested in the paper. If I understand correctly, all the LLM experiments are done using a kernel derived from representations and some kernel-based function approximator. This is not humans interact with LLMs. The paper can be greatly improved by running the exact same bandit experiment with by prompting LLMs and obtaining their behaviour. If the LLMs that have more aligned representations with humans perform better in the behavioural version of the task, the proposed hypothesis would be supported. Otherwise there is no way to know if the current bandit results translate to behaviour. In fact, just the results of such a behavioural experiment alone would be sufficient to test this hypothesis. It is unclear to me what the benefit of the kernel-based reward learning approach is. Would it not suffice to correlate reward obtained from behaviour with the representational alignment?*
>
> We appreciate the suggestion to provide a more realistic implementation of our kernel-based experiment using LLMs. To address this, we have expanded the results from our work by including a more extensive evaluation of large language models performing the text-based learning task, which we have outlined in the general response. These additional experiments provide some support to our hypothesis, though we believe that performing a full evaluation of current LLMs will require designing more challenging tasks as many existing models perform at ceiling on our current task.
>
> *> The results of the simulations are not informative. What is being shown is if you corrupt the true generative kernel that goes into the function approximator (i.e. decrease representational alignment), the model performs worse. Using the type of corruption you employ, where some scores a randomly assigned, this is to be expected under any function approximation problem with the types of models you use. I think these findings can either go into Appendix or can be removed, as they currently take up a lot of space and attention in the paper.*
>
> The goal of our simulations is to verify our theoretical results, which identify how (mis)alignment of representations transforms into performance reduction, not just the fact that it does (which is a straightforward result). The goal of this is to predict, in practice, how misalignment with human representations limits the ability to learn human values such as ethics. We show that there is a simple functional relationship between representational alignment and performance on the reinforcement learning task, consistent with the predictions of our theoretical account.
>
> In addition, we perform a reversed form of the corruption experiment in the appendix, in section A.4, “Evolution of Alignment of Language Models with Humans”. In this case, we increase the amount of representational alignment between language model kernels and human similarity judgments via interpolation, and show that there is a similar relationship between representational alignment and performance on learning human values as that observed in the simulated experiments with corruption via randomization.
>
> We appreciate the suggestion to utilize the space within the paper to further emphasize the later results, and will make edits to shorten the section accordingly.
>
> *> The conclusion drawn from the control experiment is confusing. If you mismatch the kernel with the reward function the model needs to learn, the performance goes down during personalisation. However, Table 3 in the Appendix shows that a human kernel performs better than a length kernel when the rewards are defined over length during generalisation. I appreciate the authors discuss the reasons behind this. However, the findings are followed by “[…]human-like representations are not necessarily always helpful for all tasks. Conversely, a representation based on a reward function such as the length of an action description can help support safe exploration in that particular task, but not for learning human values.” It does seem like human representations are helpful in the both tasks you defined, assuming the generalisation phase is more important than the personalisation phase.*
>
> We appreciate the reviewer’s comment. As we mention in the text below the table: “We note that the human kernel performs far better than the length kernel in generalization, both in the morality and length reward case, and the length kernel performs quite poorly on generalization for both reward functions. We note that the length kernel's performance on generalization to the length task is slightly poorer than the human kernel. After running additional experiments, we confirmed that the length kernel is highly sensitive to the choice of personalization/generalization action sets and thus performed quite poorly on a few experiments, but typically still outperforms or is comparable to the human kernel on this task. “
>
> We agree that this could be made more clear via the data presented, and we will update the table with data from multiple trials of the control experiment in the final version of the paper to better demonstrate this.

---

> ### Author Response · Authors · 2024-08-07
>
> *> More minor suggestions to improve clarity:*
>
> *- It is hard to distinguish the two green colours used in Figure 3. Your legends overlap with the y-axis labels.*
>
> *- “The representational alignment of a particular agent is measured as the Spearman correlation between the upper triangular, off-diagonal entries of the corrupted and actual similarity matrix (because diagonal entries are all the same, and the similarity matrix is symmetric).” =-- - - Pretty much the same sentence is repeated later in the text.*
>
> *- Please consider making your figure captions more informative. Figure 3 and Figure 4 almost have identical captions. Also, for Figure 3, the caption only mentions embedding models but that is not just what is plotted. Some of the sentences in the main text, such as the one starting on line 280, can be moved to the caption for better flow.*
>
> We thank the reviewer for their helpful suggestions to improve clarity and will implement these changes in the final version of the paper.
>
> *> I’m not sure what “s.t. |a| = 10.” refers to in the pseudocode.*
>
> From the algorithm description, “Randomly select $a \subset A$ s.t. $|a|=10$.” $A$ is the set of all 50 actions, and $a$ is the randomly selected set of 10 actions shown to the agent at a particular moment. $|a| = 10$ indicates that the number of allowable actions per loop is 10. We appreciate the question and have provided additional clarification on this point in the algorithm description.
>
> *> I’m confused about how actions are sampled from the function approximators’ estimates. Do you do Thompson sampling from the estimates? Pseudocode suggests so. However, in the text, Thompson sampling is described completely separately*
>
> The function approximators’ estimates provide some scalar prediction of the expected reward from each action. However, an agent relying entirely on this estimate (which is initially poor and uninformative) will not trade off between exploration and exploitation in its environment, and instead will continually take actions for which it received some nonzero reward in the beginning. We apply Thompson sampling to these scalar expected reward predictions for the agents in order to induce a smoother learning behavior, such that agents will continue to (probabilistically) explore new actions until they increase their level of certainty on which are the best actions to take. This is mentioned in one line of the pseudocode from Algorithm 1: “Choose a new action $x$ via Thompson sampling over agent's predicted rewards.” We appreciate the question and have placed additional emphasis on this point in the paper.

---

> > ### Author Response · Authors · 2024-08-12
> > **Author-Reviewer Discussion period ending**
> >
> > With the author-reviewer discussion period ending soon, we wanted to take this opportunity to thank you again for your suggestions and to check whether you had any remaining questions after our response above. We especially hope that you have a chance to review the additional experiments we ran based on your suggestions and described in the general response (official comment titled "Additional Experiments: Few-Shot Learning with LLMs"). We re-link the anonymized 1-page pdf from that general response here for convenience: https://drive.google.com/file/d/1lb0GMAbuMaiLmNwUkYpUuBZ47BjJEAFQ/view?usp=sharing
> >
> > If we've addressed your concerns, we'd be grateful if you'd consider updating your score!

---

> > ### Comment · Reviewer_7UgR · 2024-08-12
> >
> > I thank the authors for the clarifications and the additional analyses. Given the additional evidence linking the representational analyses to behavior, I am happy to raise my score to a 5.
> >
> > The reason I’m not giving a higher score is the limited scope of the evidence (ceiling effects, weak correlations, and limited models). I agree with the authors that a harder task that can provide a better test in more realistic settings would be highly beneficial in the future. It would be particularly interesting if this could be a task that all LLMs agree to respond to.

---

> ### Comment · Area_Chair_ym8x · 2024-08-12
> **Additional experiments provided past rebuttal deadline**
>
> Hi Reviewer 7UgR,
>
> ~~The authors are not allowed to provide external links to additional results. See the guidelines:~~
>
> ~~"Can we include an anonymous link in the author rebuttal? No. Do not use links in any part of the response. The only exception is if the reviewers asked for code, in which case you can send an anonymized link to the AC in an Official Comment (make sure all linked files are anonymized)." https://neurips.cc/Conferences/2024/PaperInformation/NeurIPS-FAQ~~
>
> ~~I made an exception for these authors to provide their allowed rebuttal PDF as an external link because they had some technical issue uploading the PDF. But, they are not allowed to modify it past the rebuttal deadline of August 6. Therefore, please disregard the additional results provided on August 11 when adjusting your score.~~
>
> Please disregard the above, the authors have clarified they did not modify the PDF

---

> > ### Author Response · Authors · 2024-08-12
> > **Pdf not modified since posting on Aug 7**
> >
> > Dear Area Chair,
> >
> > We have not modified the pdf on August 11. This is the exact same link and pdf as posted in our general response. By clicking details on the Google doc link you can see that the pdf has **not been modified** since being created on Aug 7, which is when we posted the anonymized link. Please let us know if you'd still like us to remove it.

---

### Official Review · Reviewer_C7Ry · 2024-07-13

**Soundness:** 4
**Presentation:** 4
**Contribution:** 3
**Rating:** 8
**Confidence:** 3

**Summary:**

This paper looks at the importance of having human-like representations for learning human values. It does this for kernel methods specifically, allowing it to operate on the level of the covariance matrix implied by the representation, rather than the representation itself. The paper presents a number of results: 1) Theoretical analysis shows that misalignment in the covariance matrix between a “teacher” (the human) and a “student” (the AI system) can result in error, and that this error is most impacted by misalignment in the matrix measuring similarity between training and test data. 2) synthetic experiments in a multi-armed bandit setting confirm the theoretical analysis; they show that increasing misalignment between representations leads to decreased performance and an increase in the number of immoral actions taken. 3) Experiments with embeddings extracted from real language models again show this negative relation between representational alignment with humans and a kernel method’s ability to learn human values. A more fine-grained version of this experiment which looks at learning individual types of human values (fairness, morality, …) confirms the existence of this relation across nearly all types.

**Strengths:**

In general this is a very solid paper which leaves little to be desired. It identifies an interesting and important question in ML: namely how important human-aligned representations are to learning human values. It contributes answers to this question from a variety of angles, including theoretical analysis and real-world empirical experiments. All experiments are to my judgement sound and report statistical significance. The writing is excellent and I appreciate the inclusion of a refresher on kernel methods in the appendix.

**Weaknesses:**

The last part of the theoretical analysis in 3.1 looks at a setting with two training examples and one test example. I think this would have been stronger if it had also included the more general setting with $N$ training examples and $M$ test examples.

**Questions:**

This work has focused on kernel methods. How much do these results now tell us about non-kernel methods? For example about estimators that can learn (near-) arbitrary functions of the given representations?

From line 182, you use variables $c_T^g$ and $c_S^g$. These have not been defined. Though one could guess from context what they mean, it would be better to introduce them properly.

As you point out in the discussion, human values vary significantly across cultural and individual levels. A reference to the Moral Machine Experiment [1], which specifically sought to quantify this variation, would help to furnish this point.

[1] Awad, E., Dsouza, S., Kim, R. *et al.* The Moral Machine experiment. *Nature* **563**, 59–64 (2018). https://doi.org/10.1038/s41586-018-0637-6

**Limitations:**

Limitations such as the representativeness of the collected data and the recognition that human judgement is varied is properly addressed in the discussion.

---

> ### Author Rebuttal · Authors · 2024-08-07
>
> *> The last part of the theoretical analysis in 3.1 looks at a setting with two training examples and one test example. I think this would have been stronger if it had also included the more general setting with N training examples and M test examples.*
>
> Thank you for the great suggestion! We have now extended the theory to the general case with n training examples and m test examples:
>
> “We can extend this result to the case where there are $n$ training examples and $m$ test examples. Let $e_m, e_n$ be column vectors consisting of $m$ and $n$ ones, respectively. To allow us to find the analytical form of the prediction expression, suppose that covariance between each pair of training examples is $c^p \neq 1$, that training examples are normalized to have variance $1$, and that the covariance between each pair of train and test examples is $c^g$. Then $K_T=(1-c^p)I + c^p e_n e_n^\top$ and $K^*_T=c^g y_m y_n^\top$. Applying the Sherman-Morrison formula and simplifying the resulting expression we get $K^{-1}_T=(1-c^p)^{-1}(I-\frac{c^p}{1+(n-1)c^p}e_n e_n^\top)$. Thus, the prediction is now $\hat{y}^g=K_T^{*\top}K_T^{-1}y^p=\frac{nc^g [1+(k-2)c^p]}{(1-c^p)[1+(k-1)c^p]}e_m e_k^\top y^p$. Misalignment in $K^*$, which can be represented by $|c^g_T-c^g_S|=\epsilon$, results in error $|\hat{y}^g-\tilde{y}^g|=|\epsilon d^p| y^p$ where $d^p$ is a function of $c^p$ but constant in $c^g$. Thus, error due to misalignment in $K^*$ grows linearly. Misalignment in $K$, which can be represented as $|c^p_T-c^p_S|=\epsilon$, results in error $|\hat{y}^g-\tilde{y}^g|=|(\frac{1}{1-c^p_T} - \frac{1}{1-c^p_T + \epsilon}d^g| y^p$ where $d^g$ is a function of $c^g$ but constant in $c^p$. Thus, error due to misalignment in $K$ ranges from $0$ to $(1-c^p_T)^{-1}d^g| y^p$ and grows sublinearly with $\epsilon$. The resulting conclusions are therefore the same as in the special case of two training examples and one test example, the error grows monotonically as representational alignment decreases and misalignment in $K^*$ has a larger effect on student performance than the same degree of misalignment in $K$ does.”
>
> *> This work has focused on kernel methods. How much do these results now tell us about non-kernel methods? For example about estimators that can learn (near-) arbitrary functions of the given representations?*
>
> We focused on kernel methods to establish our theoretical results and validate these results empirically. We appreciate the suggestion to test the generalization of our experiment to more complex estimators, and ran the same experiment using a variety of LLMs in a few-shot learning setting; results are presented in the general response.
>
> *> From line 182, you use variables cTg and cSg. These have not been defined. Though one could guess from context what they mean, it would be better to introduce them properly.*
>
> Thank you, this is a great catch. We have now clarified that these refer to the teacher and student’s beliefs, respectively, about the covariance between the training and testing examples.
>
> *> As you point out in the discussion, human values vary significantly across cultural and individual levels. A reference to the Moral Machine Experiment [1], which specifically sought to quantify this variation, would help to furnish this point.*
>
> We thank the reviewer for the helpful suggestion of an additional reference and have added it to the paper.

---

> > ### Comment · Reviewer_C7Ry · 2024-08-11
> >
> > Thank your for the response and thank you for fixing the LaTex formatting.
> >
> > I have read the reviews of the other reviewers and the authors' responses. I agree with reviewer 7UgR that a direct evaluation of the morality of LLM's outputs versus their representational alignment would provide additional support for the central hypothesis of the paper, and would strongly encourage the authors to do such an experiment. However, conference papers have limited space for experiments, and in my view the experiments that are currently in the paper provide sufficient support for the hypothesis. Therefore, as concerns the current submission, I will maintain my score.

---

> > > ### Author Response · Authors · 2024-08-12
> > > **Additional experiments in general response**
> > >
> > > Thank you for responding!
> > >
> > > Based on the suggestions from reviewer 7UgR we ended up running those additional experiments that directly evaluate morality both by having the LLMs play the bandit game, and by getting morality ratings directly from the LLMs.
> > > We summarized the additional experiments in the general response (official comment titled "Additional Experiments: Few-Shot Learning with LLMs"). We re-link the anonymized 1-page pdf from that general response here for convenience: https://drive.google.com/file/d/1lb0GMAbuMaiLmNwUkYpUuBZ47BjJEAFQ/view?usp=sharing
> > >
> > > We'd be grateful if you could check out the experiment description and results and let us know if this is what you were suggesting!
> > > If yes, we'd of course also be really grateful if you consider updating your score.

---

> ### Author Response · Authors · 2024-08-07
> **Fixed markdown LaTeX**
>
> We noticed after submitting the rebuttal that the LaTeX resulted in poor formatting, but rebuttals can no longer be edited so we are copying a fixed version here:
>
> > *The last part of the theoretical analysis in 3.1 looks at a setting with two training examples and one test example. I think this would have been stronger if it had also included the more general setting with N training examples and M test examples.*
>
> **Response**: Thank you for the great suggestion! We have now extended the theory to the general case with n training examples and m test examples:
>
> “We can extend this result to the case where there are $n$ training examples and $m$ test examples. Let $e_m, e_n$ be column vectors consisting of $m$ and $n$ ones, respectively. To allow us to find the analytical form of the prediction expression, suppose that covariance between each pair of training examples is $c^p \neq 1$, that training examples are normalized to have variance $1$, and that the covariance between each pair of train and test examples is $c^g$. Then $K_T=(1-c^p)I + c^p e_n e_n^\top$ and $K^\*_T=c^g y_m y_n^\top$. Applying the Sherman-Morrison formula and simplifying the resulting expression we get $K^{-1}_T=(1-c^p)^{-1}(I-\frac{c^p}{1+(n-1)c^p}e_n e_n^\top)$. Thus, the prediction is now $\hat{y}^g=K_T^{\*\top}K_T^{-1}y^p=\frac{nc^g [1+(k-2)c^p]}{(1-c^p)[1+(k-1)c^p]}e_m e_k^\top y^p$. Misalignment in $K^\*$, which can be represented by $|c^g_T-c^g_S|=\epsilon$, results in error $|\hat{y}^g-\tilde{y}^g|=|\epsilon d^p| y^p$ where $d^p$ is a function of $c^p$ but constant in $c^g$. Thus, error due to misalignment in $K^\*$ grows linearly. Misalignment in $K$, which can be represented as $|c^p_T-c^p_S|=\epsilon$, results in error $|\hat{y}^g-\tilde{y}^g|=|(\frac{1}{1-c^p_T} - \frac{1}{1-c^p_T + \epsilon}d^g| y^p$ where $d^g$ is a function of $c^g$ but constant in $c^p$. Thus, error due to misalignment in $K$ ranges from $0$ to $|(1-c^p_T)^{-1}d^g| y^p$ and grows sublinearly with $\epsilon$. The resulting conclusions are therefore the same as in the special case of two training examples and one test example, the error grows monotonically as representational alignment decreases and misalignment in $K^*$ has a larger effect on student performance than the same degree of misalignment in $K$ does.”

---

> ### Comment · Area_Chair_ym8x · 2024-08-12
> **Disregard additional experiments provided past the rebuttal deadline**
>
> Authors: Technically providing a link to outside results isn't allowed, you are allowed to provide a single rebuttal PDF within the time from of the author rebuttal period. Although I initially allowed it due to your comment about technical difficulties, it is not valid to provide additional results with an external link beyond the deadline of the rebuttal period, since this isn't allowed for other authors. To be fair, **please revert the PDF at the link you provided to the version that you had at the end of the rebuttal period**. I would not want to have to disqualify your paper.
>
> ~~Reviewers: please disregard these additional results.~~

---

> > ### Author Response · Authors · 2024-08-12
> > **Pdf not modified since posting on Aug 7**
> >
> > Dear Area Chair,
> >
> > We have not modified the pdf on August 11. This is the exact same link and pdf as posted in our general response. By clicking details on the Google doc link you can see that the pdf has **not been modified** since being created on Aug 7, which is when we posted the anonymized link. Please let us know if you'd still like us to remove it.

---

> ### Comment · Reviewer_C7Ry · 2024-08-13
>
> > Based on the suggestions from reviewer 7UgR we ended up running those additional experiments that directly evaluate morality both by having the LLMs play the bandit game, and by getting morality ratings directly from the LLMs. We summarized the additional experiments in the general response (official comment titled "Additional Experiments: Few-Shot Learning with LLMs"). We re-link the anonymized 1-page pdf from that general response here for convenience: https://drive.google.com/file/d/1lb0GMAbuMaiLmNwUkYpUuBZ47BjJEAFQ/view?usp=sharing
>
> Thank you for pointing this out to me. I apologize for having initially missed the additional direct evaluation of morality. Taking these results into account, I think the paper now provides solid support for the central hypothesis. I will raise my score to reflect this.

---

### Official Review · Reviewer_W1mH · 2024-07-17

**Soundness:** 2
**Presentation:** 1
**Contribution:** 2
**Rating:** 5
**Confidence:** 2

**Summary:**

The paper addresses the challenge of ensuring that machine learning models learn to achieve explicit objectives without causing harm or violating human standards, which is crucial as these models operate in more open environments. They specifically focused on value alignment in LLMs, and note that this is challenging when models must align with user preferences, values, or morals after minimal interaction.

The authors propose that learning human-like representations, or representational alignment, can aid in quickly and safely learning human values. They design a reinforcement learning task involving morally-salient actions to explore this. They collected a dataset of human value and similarity judgments to simulate AI personalization settings, and conducted a human evaluation.

**Strengths:**

1) The paper introduces the concept of representational alignment as a means to achieve value alignment, which is a relatively unexplored area in AI research.
2) The authors design a specific reinforcement learning task and create a new dataset.
3) They collect human value and similarity judgment data

**Weaknesses:**

1) The experimental evaluation is not very extensive / the motivation for this is missing and it is hard to interpret the results. The results section must be expanded more thoroughly.
2) The reinforcement learning task and dataset used in the study is a bit narrow to generalize the findings across all possible real-world scenarios.
3) While the paper focuses on safe exploration, it does not deeply explore the ethical implications of the work / discussion should be expanded here.
4) The criteria and metrics used to evaluate the success of representational alignment in achieving value alignment could be more comprehensive. More detailed metrics would help in better assessing the effectiveness of the proposed approach.

**Questions:**

1) Why is it important for LLMs to learn human-like representations to learn human values? How can we validate that the LLM is doing this  process?
2) How does this process work when there are several human values to consider?

**Limitations:**

Refer to weaknesses for more limitations.

---

> ### Author Rebuttal · Authors · 2024-08-07
>
> *> The experimental evaluation is not very extensive / the motivation for this is missing and it is hard to interpret the results. The results section must be expanded more thoroughly.*
>
> Thank you for these suggestions. We will expand the results section in the final version of the paper. We have added additional text clarifying the motivation, namely:
>
> We measure performance of agents in terms of mean reward (i.e. mean morality score), as well as number of immoral actions taken. We seek to develop learning agents who can both learn human values effectively (generalization ability) and perform their learning process in a safe, harmless manner (personalization and safe exploration), and these metrics help us to evaluate agents' performance with respect to both of these goals.
>
> In addition, we have expanded the results by including a more extensive evaluation of large language models performing the text-based learning task, as outlined in the general response.
>
> *> The reinforcement learning task and dataset used in the study is a bit narrow to generalize the findings across all possible real-world scenarios.*
>
> Thank you for this observation. We incorporated a second experiment into the paper to try to address this concern, showing that our results generalize across 10 different human value functions, which we believe improves the generalizability of our results. Our focus on the reinforcement learning task was based on the heavy use of this kind of task in the value alignment literature (e.g. [1]), and in particular, a learning framework in which the agent performing safe exploration is particularly important. Our intent wasn’t to cover all possible real-world scenarios, but to use a task that has previously been used as a metric for value alignment to demonstrate the relevance of representational alignment in this setting. We would point out that previous work has found that representational alignment can be effective in facilitating learning other few-shot settings [2], which when combined with our results suggest that this is a more general phenomenon. We have also performed additional experiments with LLMs via few-shot learning to help address this point; results are included in the general response.
>
> [1] Nahian, M., Frazier, S., Harrison, B., Riedl, M. “Training Value-Aligned Reinforcement Learning Agents Using a Normative Prior,” 2021.
>
> [2] Sucholutsky, I., Griffiths, T. “Alignment with human representations supports robust few-shot learning,” 2023.
>
> *> While the paper focuses on safe exploration, it does not deeply explore the ethical implications of the work / discussion should be expanded here.*
>
> We appreciate the reviewer’s suggestion and have expanded the Discussion and Limitations section to include the following:
>
> This work could potentially introduce another dimension to consider when working towards building more ethical AI systems that are aligned with societal values. While we hope that our study will provide a new avenue for creating safe, moral, and aligned AI systems, we acknowledge that morality is a significantly more complex and multi-faceted concept than can be captured in a small number of ratings by English-speaking internet users. Our study is intended only to highlight the importance of aligning models' internal representations with the representations of their users. Our dataset should not be used as a benchmark for determining whether models are safe or moral.
>
> *> The criteria and metrics used to evaluate the success of representational alignment in achieving value alignment could be more comprehensive. More detailed metrics would help in better assessing the effectiveness of the proposed approach.*
>
> In our simulated experiments, we studied five different metrics related to safe exploration and value alignment - namely, mean reward (mean “alignment”), number of “non-optimal” actions taken (i.e. agent did not choose the most moral action available), immoral actions taken, iterations to convergence (i.e. number of personalization iterations before the agent successfully learned the set of values), and the number of unique actions the agent had to take before it learned the values effectively. We showed in the simulations that all five metrics related to the degree of representational alignment.
>
> In our experiments using human data, we study two of these metrics - namely, mean reward and immoral actions taken - in both the personalization and generalization phases (the other metrics no longer give meaningful information because we restrict the agent to a fixed number of iterations for personalization and generalization). Once again, we show that both metrics relate to the degree of representational alignment, in both personalization and generalization.
>
> We are open to suggestions of other metrics that could be used, but our choice here was based on the kinds of measures that have been used to assess value alignment in the previous literature [1], [2]. A recent survey showed that the measures of representational alignment we adopted are widely used across cognitive science, neuroscience, and machine learning [2]. We appreciate the feedback and will provide a more detailed explanation behind our choice of metrics in the final paper.
>
> [1] Hendrycks, D., Burns, C., Basart, S., Critch, A., Li, J., Song, D., Steinhardt, J. “Aligning AI with shared human values,” 2020.
>
> [2] Sucholutsky, I., Muttenthaler, L., Weller, A., Peng, A., Bobu, A., Kim, B., Love, B., Grant, E., Groen, I., Achterberg, J., Tenenbaum, J., Collins, K., Hermann, K., Oktar, K., Greff, K., Hebart, M., Jacoby, N., Zhang, Q., Marjieh, R., Geirhos, R., Chen, S., Kornblith, S., Rane, S., Konkle, T., O'Connell, T., Unterthiner, T., Lampinen, A., Müller, K., Toneva, M., Griffiths, T. “Getting aligned on representational alignment,” 2023.

---

> ### Author Response · Authors · 2024-08-07
>
> *> Why is it important for LLMs to learn human-like representations to learn human values? How can we validate that the LLM is doing this process?*
>
> While it is certainly true that non-human representations may be better for some tasks, the task of learning human values and morals is intrinsically tied to learning things in a human way. We do acknowledge that cognitive biases may be present in some settings, and this would be a very interesting direction for future work. However, we chose to adapt action descriptions from the ETHICS dataset partly because these actions are simple and straightforward enough that the moral judgments on them were largely consistent. Modern models often do not learn human-aligned representations, and are misaligned across many domains.
>
> We can validate that LLMs are learning more human-like representations by measuring the “closeness” of the pairwise similarity matrix to humans’ that we can collect given a set of stimuli, as we do in this work. In our paper, we work with pre-trained models and their (fixed) representations, and learning happens via few-shot prompting; however, the same method can be applied during the training process of language models to quantify how much their representations become more or less human-like over time.
>
> We appreciate the reviewer’s insightful question and will provide a more explicit explanation of this in our introduction.
>
> *> How does this process work when there are several human values to consider?*
>
> Our second experiment explores the case where there are multiple values that are learned using the same representation. In this setting, we show that representational alignment is beneficial across a set of 10 different human values. Our expectation is thus that alignment will be helpful in general, even though the specific relationships that are learned to capture different human values will differ.
>
> As for aggregating multiple values to form pluralistic human value judgments, that extends beyond the scope of our work; however, this is an active research area as well. Some recent work that explores this area can be found in the following paper:
>
> Sorensen, T., Jiang, L., Hwang, J., Levine, S., Pyatkin, V., West, P., Dziri, N., Lu, X., Rao, K., Bhagavatula, C., Sap, M., Tasioulas, J., Choi, Y. Value Kaleidoscope: Engaging AI with Pluralistic Human Values, Rights, and Duties, 2024.

---

> ### Author Response · Authors · 2024-08-12
> **Author-Reviewer Discussion period ending**
>
> With the author-reviewer discussion period ending soon, we wanted to take this opportunity to thank you again for your suggestions and to check whether you had any remaining questions after our response above. We especially hope that you have a chance to review the additional experiments we ran based on reviewer suggestions and described in the general response (official comment titled "Additional Experiments: Few-Shot Learning with LLMs"). We re-link the anonymized 1-page pdf from that general response here for convenience: https://drive.google.com/file/d/1lb0GMAbuMaiLmNwUkYpUuBZ47BjJEAFQ/view?usp=sharing
>
> If we've addressed your concerns, we'd be grateful if you'd consider updating your score!

---

### Author Response · Authors · 2024-08-07
**Additional Experiments: Few-Shot Learning with LLMs**

We appreciate the helpful comments and insightful questions from the reviewers. We have expanded upon a set of experiments from appendix section A.5, where we repeat our task in a typical few-shot learning setting using 9 different LLMs. This helps to show generalizability of our results to a more realistic value-alignment setting, and provides motivation for further research that expands upon our results.

The results are summarized in the following anonymized 1-page PDF: https://drive.google.com/file/d/1lb0GMAbuMaiLmNwUkYpUuBZ47BjJEAFQ/view?usp=sharing
(We were unable to upload the PDF to the official rebuttal PDF form.)

---

### Decision · Program_Chairs · 2024-09-25

**Decision:**

Accept (poster)

**Comment:**

Reviewers unanimously agree on accepting this paper, giving ratings of (5,5,6,8), leading to a relatively high average score of 6.0.

Strengths identified:
- “this is a very solid paper which leaves little to be desired” (C7Ry)
- Idea is novel, and the problem is significant / impactful:
  - “introduces the concept of representational alignment as a means to achieve value alignment, which is a relatively unexplored area in AI research.” (W1mH)
  - “The paper presents a novel approach by linking representational alignment with value alignment, addressing a significant challenge in AI safety.” (RqUK)
  - “identifies an interesting and important question in ML: namely how important human-aligned representations are to learning human values” (C7Ry)
  - “focuses on the timely problem of personalising LLM behaviour in a safe manner.” (7UgR)
- Experimental results are convincing:
  - Results are thorough, and include theoretical results, results with bandits, and “Experiments with embeddings extracted from real language models again show this [...] relation between representational alignment with humans and a kernel method’s ability to learn human values. A more fine-grained version of this experiment which looks at learning individual types of human values (fairness, morality, …) confirms the existence of this relation across nearly all types.” (C7Ry)
  - The results are likely to be of interest to the community and impactful. E.g. “It is nice to see that the two different ways of obtaining representations from open vs. closed source models yield similar results, which can be beneficial for guiding future work.” (7UgR)
  - “All experiments are to my judgement sound and report statistical significance” (C7Ry)
- Collect human data (W1mH, 7UgR)
- well-written. “I appreciate the inclusion of a refresher on kernel methods in the appendix.” (C7Ry)

Weaknesses identified:
- Experimental evaluation is limited and hard to interpret (W1mH)
- “The theoretical analysis relies on strong assumptions, such as specific kernel functions and Gaussian process regression, which may limit the generalizability of the results.” (RqUK)
- The RL task and dataset is narrow, limiting possible implications for the real-world (W1mH)
- Reviewer 7UgR initially raised a concern that the initial experiments were insufficient to truly test “the hypothesis that increased representational alignment allows for better learning of human values”, and suggested an experiment involving having humans interact with LLMs, and tesing whether LLMs with more aligned interactions perform better in this “behavioral” version of the task.

In the rebuttal, the authors conducted the experiment suggested by reviewer 7UgR, showing a significant correlation between representational alignment and mean reward obtained in the task. This convinced reviewer 7UgR to raise their score, and also provided additional evidence to complaints about limited experimentation brought by other reviewers.

Given the positive reviews, and the potential impact of the results, my recommendation is to accept.